# Combined MS/MS-NMR Annotation Guided Discovery of *Iris lactea* var. *chinensis* Seed as a Source of Viral Neuraminidase Inhibitory Polyphenols

**DOI:** 10.3390/molecules25153383

**Published:** 2020-07-26

**Authors:** Hyun Woo Kim, Soo Sung Kim, Kyo Bin Kang, Byeol Ryu, Eunjin Park, Jungmoo Huh, Won Kyung Jeon, Hee-Sung Chae, Won Keun Oh, Jinwoong Kim, Sang Hyun Sung, Young-Won Chin

**Affiliations:** 1Research Institute of Pharmaceutical Sciences, College of Pharmacy, Seoul National University, Seoul 08826, Korea; hwkim8906@gmail.com (H.W.K.); soosung179@snu.ac.kr (S.S.K.); estrella56@snu.ac.kr (B.R.); eunjin_p@snu.ac.kr (E.P.); goodhjm112@snu.ac.kr (J.H.); chaeheesung83@gmail.com (H.-S.C.); wkoh1@snu.ac.kr (W.K.O.); jwkim@snu.ac.kr (J.K.); shsung@snu.ac.kr (S.H.S.); 2College of Pharmacy, Sookmyung Women’s University, Seoul 04310, Korea; kbkang@sookmyung.ac.kr; 3Herbal Medicine Research Division, Korea Institute of Oriental Medicine, Daejeon 34054, Korea; wkjeon@kist.re.kr; 4Convergence Research Center for Diagnosis, Treatment and Care System of Dementia, Korea Institute of Science and Technology, Seoul 02792, Korea

**Keywords:** *Iris lactea* var. *chinensis*, polyphenols, stilbene oligomers, chemical profiling, neuraminidase, MS/MS molecular networking, SMART

## Abstract

In this study, the chemical diversity of polyphenols in *Iris lactea* var*. chinensis* seeds was identified by combined MS/MS-NMR analysis. Based on the annotated chemical profile, the isolation of stilbene oligomers was conducted, and consequently, stilbene oligomers (**1**-**10**) were characterized. Of these, compounds **1** and **2** are previously undescribed stilbene dimer glycoside (**1**) and tetramer glycoside (**2**), respectively. Besides, to evaluate this plant seed as a rich source of stilbene oligomers, we quantified three stilbene oligomers of *I. lactea* var. *chinensis* seeds. The contents of three major stilbene oligomers—trans-ε-viniferin (**3**), vitisin A (**6**), and vitisin B (**9**)—in *I. lactea* var. *chinensis* seeds were quantified as 2.32 (**3**), 4.95 (**6**), and 1.64 (**9**) mg/g dry weight (DW). All the isolated compounds were tested for their inhibitory activities against influenza neuraminidase. Compound **10** was found to be active with the half maximal inhibitory concentration (IC_50_) values at 4.76 μM. Taken together, it is concluded that *I. lactea* var. *chinensis* seed is a valuable source of stilbene oligomers with a human health benefit.

## 1. Introduction

Stilbene is a class of natural polyphenols that can construct complex oligomeric structures [1]. Plant-derived stilbenes are recognized as secondary metabolites to protect host plants in the presence of exogenous stimuli or attacks, such as physical injury, UV irradiation, or pathogenic invasion [2,3,4]. The structural variety of stilbene oligomers is derived from either C–C or C–O–C bonds with two, four, six, or eight linkage points on monomeric stilbene units [5], and the stilbene oligomer formation is known to be easily mediated by oxidative couplings. These stilbene oligomers have been reported to possess many bioactivities, including anti-inflammatory, anti-microbial, antioxidant, antitumor, hepatoprotective, and metabolic disorder modulatory activities [5,6]. Stilbene monomers or dimers have been well detected in the palatable fruits, such as mulberry (*Morus alba* L.), bilberry (*Vaccinium myrtillus* L.), and blueberry (*Caccinium corymbosum* L.), and these fruits are being used as ingredients of dietary supplements or functional foods due to, in part, bioactivities of stilbenes contained in the fruits [7,8,9]. However, the stilbene oligomers are present in very small amounts in plants, and the distribution of the stilbene oligomers in the plant kingdom is not wide but seems to be limited to specific plant families like Iridaceae, Moraceae, Orchidaceae, Polygonaceae, and Vitaceae [5] despite their usage as dietary supplements or functional foods.

Iridaceae comprises 1800 species in 60 genera, representing one of the largest families of the superorder Lilianae [10]. *Iris lactea* var. *chinensis*, an herbaceous perennial plant in Iridaceae, is widely distributed in East Asia, and due to its beautiful appearance and environmental stress-tolerant traits, this plant is used in the gardening or for vegetation regeneration [11,12]. The seed of this plant has been documented as a traditional medicine in China [11]. According to the literature, this plant seed has flavonoids, xanthones, and stilbenes [11]. Thus, to assess this plant seed as a natural source containing the new or various type of stilbenes, the chemical diversity of phenolic compounds in the seeds of *I. lactea* var. *chinensis* has been explored by using the combination of LC-MS/MS-based untargeted metabolomics technique known as an MS/MS molecular networking and NMR-convolutional neural network-based structure analysis, SMART [13].

Mass spectrometry and NMR spectroscopy are the most common techniques in metabolomics studies, and each brings its strength and weakness. In mass spectrometry, the structure information is described by the size of the molecular ion and the fragmentation patterns of the molecular ions with high sensitivity. The diversity of fragmentation ion patterns belongs to molecular structures, so analysis of these patterns. However, results from mass spectrometry depend on the ionizabilities of targeted molecules. Unlike mass spectrometry, NMR spectroscopy is quantitative and produces highly reproductive results. Even the sensitivity of NMR is lower than that of mass spectrometry, NMR data is not influenced by the type of molecules. Moreover, the sensitivity and acquisition time have been improved by the development of cryo- and microprobes [14]. Thus, the complementary application of these two techniques is appropriate to study natural products that have plenty of chemical diversity.

The MS/MS molecular networking organizes MS/MS spectra into a spectral similarity-based network, which groups specialized metabolites into several molecular families based on possible structural similarity [15,16]. This molecular networking process can provide useful information depending on demands like dereplication and prioritization or discovery of a similar type of structure in bacteria, fungi, marine, and plant natural products [17,18,19].

SMART 2.0 is a convolutional neural network-based approach for the rapid annotation of molecularly diverse natural products [13,20], which has been trained on 25,000 heteronuclear single quantum coherence (HSQC) spectra from marine and terrestrial natural products and available on the website (http://smart.ucsd.edu). This tool allows characterizing the natural products by the analysis of HSQC spectroscopy, which is a general used 2D NMR method. SMART identifies previously described molecules easily and gives partial or whole structural information for undescribed molecules. 

Here, we described the combined MS/MS-NMR chemical discovery on the ethanolic extract of *I. lactea* var. *chinensis* seeds, and then based on the information generated by molecular networking and SMART analyses, further isolation and identification of two new stilbene oligomers (**1** and **2**) and eight known stilbene oligomers (**3**–**10**), followed by the quantitative analysis using three major compounds (**3**, **6**, and **9**) obtained in this study in order to evaluate the potential of *I. lactea* var. *chinensis* as the source of stilbene oligomers. Furthermore, the isolated compounds (**1**–**10**) were tested for their neuraminidase inhibitory activities [21].

## 2. Results and Discussion

### 2.1. Annotation of the Stilbenoids from I. lactea var. Chinensis Using MS/MS-NMR Analyses

To analyze chemical diversity of *I. lactea* var. *chinensis*-derived secondary metabolites, the MS/MS molecular network analysis was carried with *I. lactea* var. *chinensis* extract together with trans-ε-viniferin, which is a typical stilbenoid from natural products. The result showed three major molecular families (MFs) (Figure 1). MF A was composed of nodes with *m/z* 500–900 molecular weight ranges. Three red nodes in A were identified as procyanidin B type dimers of catechin by matching MS/MS fragmentation patterns with the GNPS (global natural products social molecular networking) spectral library. The difference of 288 Da between spectral nodes of 577 and 865 was deduced from the catechin block. MF B represented stilbene dimers, and one node was identified as *trans*-ε-viniferin (**3**) by matching MS/MS fragmentation and retention time of the standard compound. MF C was a large molecular family in which precursor ion of spectral nodes was in the range of *m/z* 600–1200. Even though none of the nodes in C were annotated by spectral library matching, precursor mass differences shown at 162 Da (hexose) or 452 Da (two stilbene moieties) inferred that spectra in MF C came from stilbene oligomers and their glycosides. 

To confirm the scaffold of MF C with the range of *m/z* 600–1200, the SMART analysis was carried out with ethyl acetate extract of *I. lactea* var. *chinensis* in which polar primary metabolites within sugars or organic acids were removed during the solvent extraction. From SMART analysis with the HSQC spectra of the fraction, 100 nearest structural neighborhoods were suggested. This result was filtered by molecular weight range of 600–1200 Da, which was the molecular weight range of MF C. The result annotated that the secondary metabolites in the MF C were stilbene oligomers, supporting the prediction from molecular network analysis (Figure 2).

Combining the results of MS/MS molecular network and SMART analysis, major chromatographic peaks were tentatively annotated in a scaffolding level (Figure 3A and Table 1). Especially fourteen peaks were annotated as stilbenoids targeted to isolate: two dimers (peaks H and I), four dimer glycosides (peaks C–F), five tetramers (peaks K, M–P), and three tetramer glycosides (peaks G, J, and L). Stilbene monomers were not observed in the result. To confirm the annotation on stilbenoids, targeted isolation and structural elucidation were conducted thereafter.

### 2.2. Structure Elucidation of Isolated Stilbenoids from I. Lactea var. Chinensis Seeds

Based on the presence of stilbenoids predicted from the MS/MS-NMR combined analyses, the EtOAc- and BuOH-soluble extracts of *I*. *lactea* var. *chinensis* seeds (Figure 4) were selected for the isolation of stilbenoids and their glycosides. As a result, two new compounds **1** and **2** were isolated along with eight known structures, which were assigned by comparison of MS and NMR data with literature, as *trans*-*ε*-viniferin (**3**) [12], *trans*-ε-viniferin-13b-β-d-glucopyranose (**4**) [5], vatalbinoside C (**5**) [22], vitisin A (**6**) [23], *cis*-vitisin B (**7**) [24], *cis*-vitisin C (**8**) [25], vitisin B (**9**) [23], and vitisin C (**10**) [23] (Appendix A). 

Compound **1** was obtained as a brown amorphous powder. Its molecular formula is C_40_H_42_O_16_, which was confirmed by the analysis of the high resolution electrospray ionization mass spectrometry (HRESIMS) data (C_40_H_41_O_16_, *m/z* 777.2381 [M−H]^−^, calcd for 777.2395). The ^1^H-NMR data of **1** showed aromatic signals for two 4-hydroxybenzene groups at δ_H_ 6.96 (2H, d, *J* = 8.6 Hz, H-2a, H-6a), 6.73 (2H, d, *J* = 8.6 Hz, H-3a, H-5a), 6.92 (2H, d*, J* = 8.6 Hz, H-2b, H-6b), 6.60 (2H, d, *J* = 8.6 Hz, H-3b, H-5b), and two aliphatic protons of a dihydrobenzofuran ring at δ_H_ 5.24 (1H, d, *J* = 6.1 Hz, H-7a) and 3.33 (1H, d, *J* = 6.1 Hz, H-8a). The presence of one 3,5-dihydroxybenzene moiety was also deduced from the proton signals at δ_H_ 6.41 (1H, t, *J* = 2.1 Hz, H-12a), 6.22 (1H, brs, H-10a), and 6.12 (1H, brs, H-14a) (Table 2 and Appendix A). The coupling constant (12.0 Hz) of H-7b and H-8b suggested the presence of a *cis*-olefinic double bond. These results showed that the proton signals of compound 1 were similar to those of *cis*-viniferin, which is composed of two stilbene units with the formation of dihydrobenzofuran. The heteronuclear multiple bond correlation (HMBC) correlations of δ_H_ 5.24 (H-7a) with δ_C_ 128.5 (C-2a) and 147.0 (C-9a) and the correlations of δ_H_ 6.25 (H-7b) with δ_C_ 131.2 (C-2b) and 137.5 (C-9b) also supported the presence of *cis*-viniferin structure. Besides, two anomeric proton signals at δ_H_ 4.73 (1H, d, *J* = 7.1 Hz, H-1′) and 4.76 (1H, d, *J* = 7.1 Hz, H-1″) showed the presence of two sugar units. The locations of sugar units were confirmed as C-11a and C-13b by the HMBC correlations of δ_H_ 4.73 (H-1′) with δ_C_ 160.4 (C-13b) and δ_H_ 4.76 (H-1″) with δ_C_ 160.3 (C-11a) (Appendix A). Each sugar unit was identified as D-glucose (Appendix A) by acidic hydrolysis of 1 and the subsequent conversion to an arylthiocarbamoyl-thiazolidine derivative by reaction with l-cysteine and *o-*tolyl isothiocyanate [26]. To confirm the relative configuration of C-7a and C-8a, the coupling constants of H-7a and H-8a were compared with the literature data [27] in which the coupling constant of *trans* is around 5 to 6 Hz, and that of *cis* is around 9 to 10 Hz. Therefore, the relative configuration was identified as *trans*. ECD spectrum analysis was followed to determine the absolute configuration of 1 by comparing it with that of *trans*-ε-viniferin (3). Both of them showed two strong positive Cotton effects between 200 and 260 nm (Appendix A), which suggested that the absolute configuration of 1 was 7a*R* and 8a*R*. Consequently, compound 1 was identified as *cis*-ε-viniferin-11a,13b-di-*O*-β-d-glucopyranoside.

Compound **2** was isolated as a brown amorphous powder, and HRESIMS data in the negative mode determined its molecular formula as C_62_H_52_O_17_ (C_62_H_51_O_17_, *m/z* 1067.3105 [M−H]^−^, calcd for 1067.3126). The formula suggested that compound 2 was stilbene tetramer with one hexose unit. The ^1^H-NMR spectrum (Appendix A) of 2 showed the presence of three 4-hydroxybenzene moieties with aromatic proton signals at δ_H_ 7.00 (2H, d, *J* = 8.6 Hz, H-2a, h-6a), 6.74 (2H, d, *J* = 8.6 Hz. H-3a, H-5a), 6.60 (2H, d, *J* = 8.6 Hz, H-2c, H-6c), 6.55 (2H, d, *J* = 8.6 Hz, H-3c, H-5c), 7.12 (2H, d, *J* = 8.4 Hz, H-2d, H-6d), and 6.77 (2H, d, *J* = 8.4 Hz, H-3d, H-5d). Furthermore, two characteristic triplet-like aromatic proton signals at δ_H_ 6.10 (1H, t, *J* = 2.1 Hz, H-12a) and 6.05 (1H, t, *J* = 2.1 Hz, H-12d) indicated that two 3,5-dihydroxybenzene groups existed. From six doublet proton signals at δ_H_ 5.26 (1H, d, *J* = 6.5 Hz, H-7a), 3.91 (1H, d, *J* = 6.5 Hz, H-8a), 5.44 (1H, d, *J* = 5.8 Hz, H-7c), 4.22 (1H, d, *J* = 5.8 Hz, H-8c), 5.30 (1H, d, *J* = 5.2 Hz, H-7d), and 4.26 (1H, d, *J* = 5.2 Hz, H-8d), the presence of three dihydrobenzofuran groups was suggested. These results showed that the proton signals of compound 2 were similar to those of cis-vitisin B (7). One anomeric proton signal at δ_H_ 4.72 (1H, d, *J* = 7.2 Hz, H-1′) announced that one sugar unit was substituted, and the location of sugar unit was confirmed on the C-13b by the HMBC correlation of δ_H_ 4.72 (H-1′) with δ_C_ 160.0 (C-13b) (Appendix A). The sugar unit was identified as D-glucose by the same method of 1 (Appendix A). The absolute configuration of the aglycone part in compound 2 was determined to be the same as cis-vitisin B (7) by observing that the ECD curves of compound 2 and cis-vitisin B (7) are superimposable. (Appendix A). Thus, compound 2 was determined as cis-vitisin B-13b-*O*-β-d-glucopyranoside.

### 2.3. Quantification of Stilbenoids in I. lactea var. Chinensis Seed Extract

The contents of stilbene oligomers in *I. lactea* var. *chinensis* seeds were quantified by using HPLC-UV. From HPLC-UV chromatogram of *I. lactea* var. *chinensis* extract, three major stilbene peaks were chosen for the quantification of their contents in the extract (Appendix A). The representative HPLC-UV chromatogram is depicted in Figure 3B. The analysis method was validated by recoveries and reproducibility against the three constituents. The quantitative HPLC-UV analysis of *trans*-ε-viniferin (3), vitisin A (6), and vitisin B (9) was carried out by the external calibration curves of the isolated standards (Table 3). 

The results showed good linearity between the peak area and concentration. All calibration curves were linear with correlation coefficients (R2) of over 0.9999. Recoveries of the three compounds were tested by the analysis of the *I. lactea* var. *chinensis* seed extract spiked with the major compounds. As shown in Table 3, the average recoveries of trans-ε-viniferin (**3**), vitisin A (**6**), and vitisin B (**9**) were 96.7%, 92.7%, and 100.8%, respectively. Precision was evaluated by inter- and intra-day analytical precision for an ethanolic crude extract of *I. lactea* var. *chinensis* seed at the concentration of 3.33 mg/mL (Table 4). Both intra-day and inter-day precisions were found to be within accepted criteria with the relative standard deviation (RSD) (<3%). From HPLC-UV analysis with the validated method, the contents of trans-ε-viniferin (**3**), vitisin A (**6**), and vitisin B (**9**) in *I. lactea* var. *chinensis* seed extract were quantified as 2.32 ± 0.06 (**3**), 4.95 ± 0.14 (**6**), and 1.64 ± 0.01 (**9**) mg/g DW (dry weight), respectively (Table 4). The stilbene contents of *I. lactea* var. *chinensis* seeds suggested that this natural product could be considered as a great source of stilbene oligomers with over 8 mg of stilbene oligomers per 1 g of the seeds. This stilbene contents were similar to those of *Vitis vinifera* (7.9 mg/g), which is known to be a stilbene-rich plant [28], but parts of the source are much sustainable in *I. lactea* var. *chinensis* seed than *Vitis* species in which most of the stilbene oligomers have been reported from the stems or canes [28,29].

### 2.4. Viral Neuraminidase Inhibitory Effects of Isolated Stilbenoids from I. lactea var. Chinensis Seeds

The viral neuraminidase inhibition assay was carried out to evaluate neuraminidase inhibitory activities of isolated compounds (1–10) against neuraminidase from the H1N1 influenza virus (Appendix A). Tamiflu (Oseltamivir), a well-known flu therapeutic molecule and a neuraminidase inhibitor [30], was used as the positive control. Among them, compounds 9 and 10, which are stilbene tetramers, showed the inhibitory activities, while compound 2, the tetramer glycoside, did not. This indicated that the glycosylation attenuated the neuraminidase inhibitory activities of stilbene oligomers. In addition, the trans type tetramers, such as compounds 9 and 10, showed stronger inhibitory activities than the cis type tetramers (7 and 8). Further, a dose-dependent response evaluation for vitisin C (10) was conducted, and as shown in Appendix A, vitisin C (10) had an IC_50_ value of 4.76 μM, while vitisin B (9) was degraded during storage, and thus further evaluation for this compound was not conducted at this time. However, a previous study reported that vitisin B (**9**) from *Vitis amurensis* inhibited viral neuraminidase activity by non-competitive inhibition [31]. In addition to the previous results, we could observe that the inhibitory activity of stilbene oligomers against viral neuraminidase deemed to be related to not only the level of oligomerization but also cis-trans isomerism.

## 3. Materials and Methods

### 3.1. General Experimental Procedure

Optical rotation was recorded on a JASCO P-2000 polarimeter (JASCO, Easton, MD, USA). UV and ECD spectra were obtained using a Chirascan and ECD spectrometer (Applied photophysics, Surrey, UK). NMR spectra were recorded on AVANCE-600 and AVANCE III HD (Bruker, Billerica, MA, USA) at 25 °C with a cryogenic probe. All HRESIMS data were measured on a Waters Xevo G2 QTOF mass spectrometer (Waters Co., Manchester, UK). Column chromatography was performed with Sephadex LH-20 (25–100 μm, Pharmacia, Piscataway, NJ, USA). TLC was carried out with Kieselgel 50 F254 coated normal silica gel TLC plate (Merck, Darmstadt, Germany). MPLC was carried out by Grace Reveleris MPLC system (Grace, IL, USA) with the C18 column (120 g, Grace, IL, USA). The preparative HPLC system was equipped with a G-321 pump (Gilson, Middleton, WI, USA), a G-151 UV detector (Gilson, WI, USA), and the Kintex C18 column (250 mm × 10 mm i.d.; 5 μm, Phenomenex, CA, USA). The analytical HPLC system was UltiMate™ 3000 equipped with UltiMate™ 3000 variable wavelength UV detector (Thermo Scientific, CA, USA) and YMC Triart C18 column (250 mm × 4.6 mm i.d.; 5 μm, YMC Co. Ltd., Japan). All solvents were purchased from Daejung Chemicals & Metals Co. Ltd. (Si-Heung, Korea). The reagents for aldose discrimination (L-cysteine methyl ester hydrochloride, ο-tolyl isothiocyanate) were purchased from Tokyo Chemical Industry CO. Ltd. (Tokyo, Japan).

### 3.2. Plant Materials

Dried seeds of *Iris lactea var. chinensis* were purchased from the oriental herb market, Health Wisdom (https://www.healthwisdom.shop). The dried seeds (Appendix A), of which the moisture content was 9.1%, used in the present study (SNU-1609), were identified by H.-S. Chae, along with DNA-based authentication (Appendix A) performed by Life Sciences Research Institute, Biomedic Co., Ltd. in South Korea, and deposited at the Medicinal Plant Garden, College of Pharmacy, Seoul National University.

### 3.3. Sample Extraction

The air-dried seeds of *I. lactea* var. *chinensis* (2 kg) were grounded and macerated with 70% (*v*/*v*) ethanol in water (3.0 L) at room temperature for 2 days. This was repeated three times. The filtrate was concentrated in vacuo and lyophilized to yield a brown solid (50.0 g, yield: 2.5%). The extract was ground to powder and stored at 4 °C in refrigerator.

This crude extract powder (50.0 g) was suspended in water (1.5 L) and successively extracted with *n*-hexane (3 × 1.5 L, 2.9 g), ethyl acetate (3 × 1.5 L, 20.0 g), and n-butanol (3 × 1.5 L, 7.1 g), sequentially. Each extract was concentrated in vacuo and stored at 4 °C in refrigerator. 

### 3.4. LC-MS/MS-Based Metabolite Profiling

#### 3.4.1. Sample Preparation

Lyophilized crude extract of *I*. *lactea* var. *chinensis* was dissolved in LC-MS grade MeOH with sonication and filtered with a 0.2 μm PVDF membrane syringe filter (Pall Gelman Sciences, Ann Arbor, MI) with a concentration of 2 mg/mL.

#### 3.4.2. Data Acquisition and Molecular Networking Analysis

UPLC-QTOF/MS data were acquired from the Waters Acquity UPLC system (Waters Co., Milford, MA, USA), which consists of a binary solvent delivery system, autosampler, and photodiode array (PDA) detector. UPLC column was the Waters Acquity UPLC BEH C18 (150 mm × 2.1 mm, 1.7 μm). The mobile phase was 20 mM formic acid in water (A) and acetonitrile (B), with the following gradient: 10–90% B (0–14 min, *v*/*v*). The flow rate was set at 300 μL/min, and the injection volume was 2.0 μL. The temperatures of the autosampler and column oven were maintained at 15 °C and 40 °C, respectively. The MS experiments were performed on a Waters Xevo G2 QTOF mass spectrometer (Waters MS Technologies Manchester, UK) connected to the UPLC system through an electrospray ionization (ESI) interface. The ESI condition was set as follows: negative ion mode, capillary voltage 2.5 kV, cone voltage 40 V, source temperature 120 °C, desolvation gas temperature 350 °C, cone gas flow 50 L/h, and desolvation gas flow 800 L/h. The ion acquisition rate was 0.2 s. Data were centroided during acquisition using independent reference lock-mass ion via the LockSpray^TM^ interface to ensure accuracy and precision. Leucine enkephalin (*m*/*z* 554.2615 in negative mode) was used at the concentration of 200 pg/μL with an infusion flow rate of 5 μL/min.

LC-MS/MS data were analyzed using a feature-based molecular networking workflow, which is available on the GNPS web platform (https://gnps.ucsd.edu) with a spectral preprocessing by MZmine2 software [32]. All the result and parameter can be accessed with the GNPS job id for molecular network analysis (https://gnps.ucsd.edu/ProteoSAFe/status.jsp?task=b5df6b52fad649f2bceba1364a9bdbe5) and enhanced analysis (https://gnps.ucsd.edu/ProteoSAFe/status.jsp?task=34940dd84d14465184a4501f0d55808d).

### 3.5. SMART Analysis

The HSQC spectrum of the ethyl acetate extract of *I. lactea* var. *chinensis* seeds (20 mg/mL) was preprocessed using MestReNova, phase-corrected, and referenced to the solvent peak (CDCl_3_). Peak lists were obtained automatically using the embedded peak picking method. The peak list table was subjected to SMART 2.0 webserver (http://smart.ucsd.edu) and automatically converted to a 100 by 100 binary image for convolutional neural networks. Feature extraction of the submitted HSQC spectrum was applied by the SMART 2.0 pre-trained model, and 180-dimensional coordinates were given. The cosine scoring measured the similarity between the result and the compounds from the SMART database containing 25,434 experimental and 27,642 calculated ^1^H-^13^C HSQC spectra.

### 3.6. Isolation of Stilbene Oligomers

The ethyl acetate extract (20.0 g) was fractionated with a reversed-phase MPLC with a MeCN-H_2_O gradient system (30:70–100:0, *v/v*) at a flow rate of 80 mL/min to yield four fractions (EA1–EA4). The EA2 fraction was applied to Sephadex LH-20 with MeOH to yield two subfractions (EA2-1, EA2-2). The resulting EA2-2 fraction was compound **1** (7.1 mg), and the EA-2-1 fraction was delivered to a reversed-phase semi-preparative HPLC with MeCN-H_2_O isocratic system (29:71, *v/v*) at a flow rate of 4 mL/min to yield compounds **2** (3.9 mg) and 4 (11.1 mg). Compounds **3** (340 mg) and **6** (410 mg) were isolated from EA3 fraction with a reversed-phase MPLC with a MeCN-H_2_O isocratic system (45:55, *v/v*) at a flow rate of 40 mL/min. EA4 fraction was separated into two fractions (EA4-1, EA4-2) by preparative HPLC with MeCN-H_2_O isocratic system (30:70, *v/v*) at a flow rate of 16 mL/min. Compounds 7 (4.3 mg) and 8 (1.7 mg) were isolated from EA4-1 by semi-preparative HPLC with MeCN-H_2_O isocratic system (25:75, *v/v*) at a flow rate of 4 mL/min. EA4-2 fraction was delivered to a reversed-phase semi-preparative HPLC eluted with MeCN-H_2_O isocratic system (35:65, *v/v*) at a flow rate of 4 mL/min to yield compounds 9 (120 mg) and 10 (24 mg) (Appendix A).

The n-butanol extract (7.1 g) was fractionated into two fractions (BU1, BU2) by a reversed-phase MPLC with the MeCN-H_2_O gradient system (30:70–90:10, *v/v*) at a flow rate of 60 mL/min, and compound 5 (28.3 mg) was crystallized from BU1 subfraction (Appendix A).

*cis*-ε-viniferin-11a,13b-di-*O*-β-d-glucopyranoside (1): Brown amorphous solid; ${\left[\alpha \right]}_{D}^{20}$+ 61.2 (c 0.1, MeOH); UV (MeOH) λ_max_ (log ε) 230 (sh), 204 (4.7), 287 (4.2) nm; ECD (MeOH) λ_max_ (Δε) 206 (20.5), 236 (15.1), 311 (−2.9) nm; NMR data, Table 2; HRESIMS *m/z* 777.2381 [M–H]^−^ (calcd for C_40_H_41_O_16_, 777.2395).

*cis*-vitisin B-13b-*O*-β-d-glucopyranoside (2): Brown amorphous solid; ${\left[\alpha \right]}_{D}^{20}$− 18.1 (c 0.1, MeOH); UV (MeOH) λ_max_ (log ε) 225 (sh), 208 (4.9), 285 (4.3) nm; ECD (MeOH) λ_max_ (Δε) 208 (11.8), 222 (−11.2), 295 (−5.7) nm; NMR data, Table 2; HRESIMS *m/z* 1067.3105 [M − H]^−^ (calcd for C_62_H_51_O_17_, 1067.3126).

### 3.7. Quantification of Major Stilbene Oligomers Using HPLC-UV Analysis

#### 3.7.1. Preparation of Standard and Sample Solution

Three isolated and purified compounds—*trans*-ε-viniferin (**3**), vitisin A (**6**), and vitisin B (**9**)—were used as reference compounds. Stock solutions (10 mg/mL) were prepared by precisely weighing and dissolving each in 50% (*v/v*) aqueous methanol before every experiment day. A mixed standard stock solution was prepared by adding a precise volume of each stock solution to make a solution containing 1 mg/mL of each analyte. The mixed solution was diluted via serial dilution. The crude extract power (100 mg) was precisely weighed and dissolved in 50% (*v/v*) aqueous methanol with sonication before every experiment day, and the solution was diluted to sample solution concentration (3.33 mg/mL). All solutions were filtered through a 0.2 μm PVDF membrane filter before injection.

#### 3.7.2. HPLC-UV Chromatographic System

*I. lactea* var. *chinensis* seeds extract was quantitatively analyzed by HPLC-UV using the UltiMate™ 3000 system (Thermo Scientific, USA) equipped with a binary pump, an auto-sampler, and a photodiode array detector. The analytical column was a YMC Triart C18 column (250 mm × 4.6 mm i.d.; 5 μm, YMC Co. Ltd., Japan), and the mobile phase consisted of 0.1% formic acid in water (A) and acetonitrile (B), with the following gradient: 20–80% B (0–30 min). The flow rate was set at 1 mL/min, and the injection volume was 5 μL. The temperatures of the autosampler and column oven were maintained at 15 °C and 30 °C, respectively. The detection wavelength was set at 280 nm. The results were expressed as milligrams per gram of the crude extract (mg/g).

#### 3.7.3. Method Validation

The HPLC method was validated by the linearity, limit of detection and quantification, accuracy, precision, and repeatability. The linearity of the method was assessed from the correlation coefficients (R2) of the regression curves obtained for each standard. The limits of detection (LOD) and limits of quantification (LOQ) were calculated from the standard deviation of y-intercepts (σ) and the slopes (S), according to the following formulas: LOD = 3.3 × σ/S and LOQ = 10 × σ/S. Accuracy was evaluated by spiking *I. lactea* var. *chinensis* extract (3.33 mg/mL), with three concentrations of the quantified standard compounds. Precision (intra- and inter-day) of the analysis was accomplished by analyzing *I. lactea* var. *chinensis* extract (3.33 mg/mL) 3-fold on three consecutive days.

### 3.8. Evaluation of Viral Neuraminidase Inhibitory Activity

The viral neuraminidase activity was screened using 2´-(4-methylumbelliferyl)-α-D-*N*- acetylneuraminic acid (4-MU-NANA) as the fluorescent substrate. DMSO solvent was used to dilute the concentrations of all compounds and corresponding concentrations in the enzyme buffer MES (32.5 mM 2-(*N*-morpholino) ethanesulfonic acid, 4 mM CaCl_2_ pH 6.5). The assay was operated in 96-well plates containing 10 μL of virus suspensions (containing active influenza neuraminidase) and 10 μL of screening compounds. After incubation of 30 min at 37 °C, 30 μL of 4-MU-NANA substrate in enzyme buffer was added to each well. Then, two hours of an enzyme reaction was carried out in the incubation, and the stop solution (25% EtOH (*v/v*), 0.1 M glycine, pH 10.7) was added to finish the reaction. The fluorescence intensity of the reaction solution was measured with an excitation wavelength of 360 nm and an emission wavelength of 440 nm. Oseltamivir was used as a positive control, and the results are expressed as the means ± SD of triplicate experiments. Differences between group mean values were determined by one-way analysis of variance, followed by a two-tailed Student’s t-test for unpaired samples, assuming equal variances.

## 4. Conclusions

Previously, combined analyses between MS/MS and NMR were difficult because each method was used in different stages in natural product research; LC-MS was used in the early stage to dereplicate secondary metabolites, and NMR was used in the last stage to elucidate molecular structures. However, the SMART analysis allowed us to use NMR analysis in early-stage within extract or fraction levels to annotate what kinds of metabolites inside. Using both analyses together, fourteen stilbenoids were rapidly annotated, and ten stilbenoids (**1**–**10**) were identified by further isolation and purification studies. 

To evaluate the value of *I. lactea* var*. chinensis* seeds as a source of stilbene oligomers, contents of three major stilbene oligomers—trans-ε-viniferin (**3**), vitisin A (**6**), and vitisin B (**9**)—were quantified by HPLC-UV analysis, which was validated by method validation. The stilbene contents of *I. lactea* var*. chinensis* seeds suggested that this natural product could be considered as a great source of stilbene oligomers with over 8 mg of stilbene oligomers per 1 g of the seeds. All the isolates were evaluated for their viral neuraminidase inhibitory activities, and compounds **7**–**10** showed moderate inhibitory activities. From these results, *I. lactea* var*. chinensis* seed extract might be proposed as a sustainable and rich source of stilbene oligomers with viral neuraminidase inhibitory effect.

## Figures and Tables

**Figure 1 molecules-25-03383-f001:**
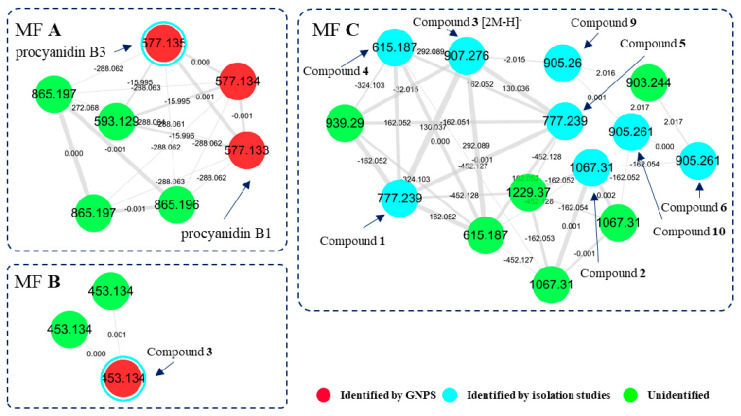
The MS/MS molecular network built from the LC-MS/MS data of the ethanolic extract of *I. lactea* var. *chinensis* seeds. Molecular families (MF) **A** (procyanidins), **B** (stilbene dimers), and **C** (stilbene oligomers) were annotated based on the spectral library matching and manual inspection on MS/MS spectra. Properties of *m/z* and mass difference were tagged on each of the nodes and edges.

**Figure 2 molecules-25-03383-f002:**
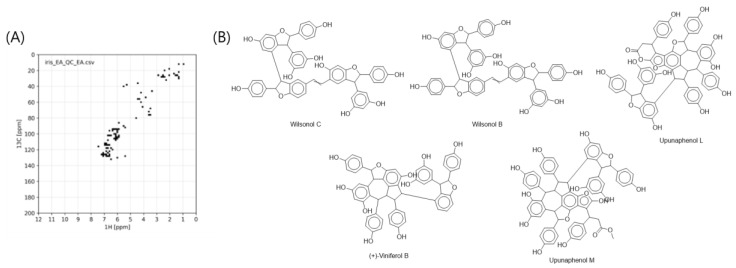
Digitized HSQC (heteronuclear single quantum coherence) spectra of ethyl acetate extracts of *I. lactea* (**A**) and their top 5 annotated structures from SMART analysis (**B**). Annotated structures were filtered by the molecular weight range of 600–1200 Da.

**Figure 3 molecules-25-03383-f003:**
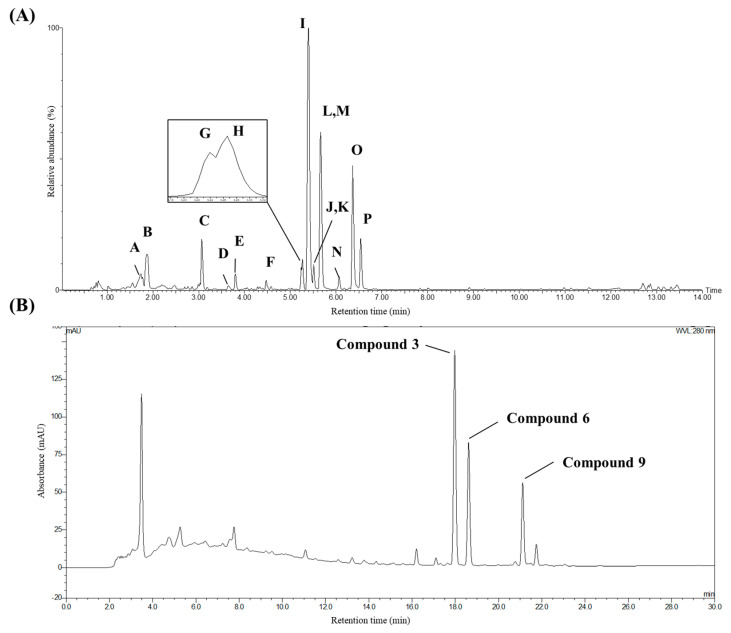
Base peak chromatogram (**A**) and HPLC-UV chromatogram (**B**) of ethanolic extract of *I*. *lactea* var*. chinensis* seeds.

**Figure 4 molecules-25-03383-f004:**
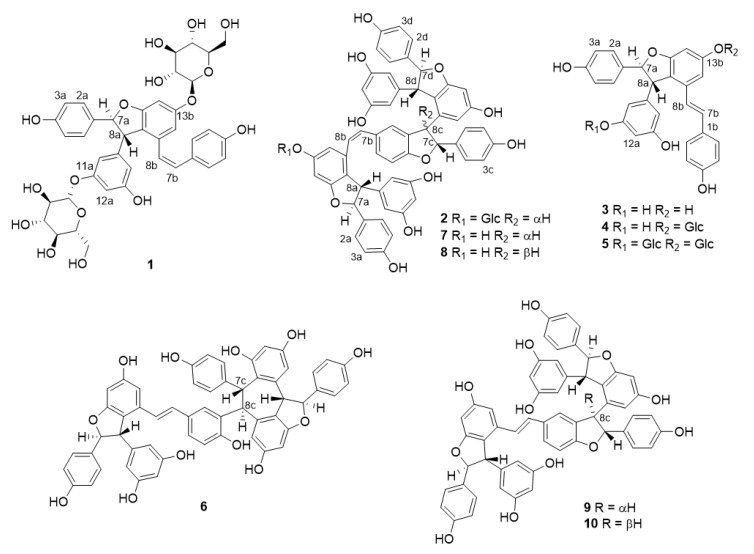
Structures of isolated stilbene oligomers from *I. lactea* var. *chinensis*.

**Table 1 molecules-25-03383-t001:** Major chromatographic peaks in the LC-MS/MS profile of the *I. lactea* var*. chinensis* seeds extract.

Peak	Retention Time (min)	Precursor Ion *m/z* ([M−H]^-^)	Molecular Formula	Error (ppm)	MS/MS Fragments	Compounds
A	1.74	577.1344	C_30_H_26_O_12_	−0.3	425, 407, 289, 125	procyanidin B3
B	1.86	289.0701	C_15_H_14_O_6_	−3.8	245, 203, 125	catechin
C	3.06	777.2402	C_40_H_42_O_16_	0.9	615, 453, 359, 347	vatalbinoside C (**5**)
D	3.64	777.2381	C_40_H_42_O_16_	−1.8	633, 575, 453	*cis*-ε-viniferin-11a,13b-*O*-β-d-diglucopyranoside (**1**)
E	3.80	615.1884	C_34_H_32_O_11_	2.9	475, 453, 359, 347	*trans*-ε-viniferin-13b-*O*-β-d-glucopyranose (**4**)
F	4.48	615.1891	C_34_H_32_O_11_	4.1	537, 475, 453, 435, 359, 347	Stilbene dimer glycoside
G	5.24	1067.3105	C_62_H_52_O_17_	1.3	905, 799, 663, 573, 453, 359, 347, 253	*cis*-vitisin B-13b-*O*-β-d-glucopyranoside (**2**)
H	5.27	453.1336	C_28_H_22_O_6_	−0.4	435, 411, 359, 347, 253, 225	Stilbene dimer
I	5.40	453.1335	C_28_H_22_O_6_	−0.7	435, 411, 385, 369, 359, 347, 253, 225	*trans*-ε-viniferin (**3**)
J	5.50	1067.3103	C_62_H_52_O_17_	1.3	905, 799, 663, 573, 453, 359, 347, 253	Stilbene tetramer glycoside
K	5.51	905.2580	C_56_H_42_O_12_	−2.0	811, 799, 453, 359, 347	*cis*-vitisin B (**7**)
L	5.63	1067.3110	C_62_H_52_O_17_	1.5	905, 799, 663, 573, 453, 359, 347, 253	Stilbene tetramer glycoside
M	5.66	905.2591	C_56_H_42_O_12_	−0.8	811, 799, 675, 545, 451, 439, 359, 347, 333	vitisin A (**6**)
N	6.06	905.2562	C_56_H_42_O_12_	−4.0	811, 799, 679, 573, 545, 477, 451, 359, 347	*cis*-vitisin C (**8**)
O	6.36	905.2609	C_56_H_42_O_12_	1.2	811, 799, 693, 545, 359, 347	vitisin B (**9**)
P	6.54	905.2591	C_56_H_42_O_12_	−0.8	811, 799, 693, 545, 451, 439, 359, 347, 333	vitisin C (**10**)

**Table 2 molecules-25-03383-t002:** ^1^H- (600 MHz) and ^13^C-NMR (150 MHz) spectroscopic data of compounds **1** and **2** in MeOH-*d_4._*

	1		2	
Position	δ_C_	δ_H_ (*J* in Hz)	δ_C_	δ_H_ (*J* in Hz)
1a	133.5		133.5	
2a	128.5	6.96 (d, 8.6)	128.5	7.00 (d, 8.6)
3a	116.3	6.73 (d, 8.6)	116.3	6.74 (d, 8.6)
4a	158.5		158.6	
5a	116.3	6.73 (d, 8.6)	116.3	6.74 (d, 8.6)
6a	128.5	6.96 (d, 8.6)	128.5	7.00 (d, 8.6)
7a	94.9	5.24 (d, 6.1)	95.2	5.26 (d, 6.5)
8a	57.6	3.33 (d, 6.1)	57.9	3.91 (d, 6.5)
9a	147.0		146.6	
10a	108.4	6.22 (brs)	107.4	5.92 (d, 2.2)
11a	160.3		159.6	
12a	103.4	6.41 (t, 2.1)	102.2	6.10 (t, 2.1)
13a	159.6		159.6	
14a	110.0	6.12 (brs)	107.4	5.92 (d, 2.2)
1b	130.0		131.5	
2b	131.2	6.92 (d, 8.6)	126.8	6.51 (m)
3b	116.1	6.60 (d, 8.6)	127.9	
4b	157.9		159.6	
5b	116.1	6.60 (d, 8.6)	110.0	6.56 (d, 8.3)
6b	131.2	6.92 (d, 8.6)	130.2	6.91 (dd, 8.3, 1.3)
7b	132.2	6.25 (d, 12.0)	131.8	6.08 (d, 12.2)
8b	126.4	6.07 (d, 12.0)	126.3	5.96 (d, 12.2)
9b	137.5		137.6	
10b	123.2		123.3	
11b	162.6		162.6	
12b	98.3	6.54 (d, 1.7)	98.1	6.47 (d, 2.1)
13b	160.4		160.0	
14b	110.6	6.55 (d, 1.7)	110.8	6.48 (d, 2.1)
13b-Glc				
1′	102.4	4.73 (d, 7.1)	102.4	4.72 (d, 7.2)
2′	74.8	3.40–3.50 (m)	74.9	3.30–3.50 (m)
3′	77.9	3.40–3.50 (m)	77.8	3.30–3.50 (m)
4′	71.1	3.40–3.50 (m)	71.0	3.30–3.50 (m)
5′	78.0	3.40–3.50 (m)	77.8	3.30–3.50 (m)
6′	62.3	3.81 (dd, 12.1, 2.2) 3.70 (dd, 12.1, 5.1)	62.2	3.78 (dd, 12.2, 2.3) 3.70 (dd, 12.2, 4.8)
11a-Glc				
1′′	102.6	4.76 (d, 7.1)		
2′′	74.8	3.40–3.50 (m)		
3′′	77.8	3.40–3.50 (m)		
4′′	71.1	3.40–3.50 (m)		
5′′	78.0	3.40–3.50 (m)		
6′′	62.2	3.84 (dd, 12.1, 2.2) 3.65 (dd, 12.1, 5.1)		
1c			132.6	
2c			127.9	6.60 (d, 8.6)
3c			116.2	6.55 (d, 8.6)
4c			158.0	
5c			116.2	6.55 (d, 8.6)
6c			127.9	6.60 (d, 8.6)
7c			92.5	5.44 (d, 5.8)
8c			52.9	4.22 (d, 5.8)
9c			142.3	
10c			120.4	
11c			162.5	
12c			96.8	6.30 (t, 2.1)
13c			160.3	
14c			107.5	6.13 (d, 2.1)
1d			134.2	
2d			128.0	7.12 (d, 8.4)
3d			116.5	6.77 (d, 8.4)
4d			158.5	
5d			116.5	6.77 (d, 8.4)
6d			128.0	7.12 (d, 8.4)
7d			95.0	5.30 (d, 5.2)
8d			57.8	4.26 (d, 5.2)
9d			147.6	
10d			107.2	5.97 (brs)
11d			159.9	
12d			102.5	6.05 (t, 2.1)
13d			159.9	
14d			107.2	5.97 (brs)

**Table 3 molecules-25-03383-t003:** Calibration data and percent of recovery rates (Rec, high, medium, and low spike) for three major compounds in *I*. *lactea* var. *chinensis* seeds, including regression equation, correlation coefficient (R^2^), and limit of detection and quantitation (LOD and LOQ; values in mg/mL).

Compound	Regression Equation	R^2^	LOD	LOQ	Rec 1 (High)	Rec 2 (Medium)	Rec 3 (Low)
**3**	y = 73.868*x* + 0.4495	0.9999	0.0009	0.0028	95.9	96.4	97.6
**6**	y = 17.892*x* + 0.4495	1.0000	0.0040	0.0122	90.4	92.7	94.9
**9**	y = 42.076*x* + 0.4495	0.9999	0.0024	0.0073	100.9	100.5	100.9

**Table 4 molecules-25-03383-t004:** Intra- and Inter-day precision of analysis and the contents of three major compounds in *I. lactea* var. *chinensis* seeds.

Compound	Intra-Day (*n* = 3) ^a^			Inter-Day (*n* = 3) ^a^	Contents (mg/g DW)
	Day 1	Day 2	Day 3		
**3**	21.19 (2.1)	21.56 (0.5)	21.60 (0.4)	21.45 (1.1)	2.32 ± 0.06
**6**	10.76 (2.8)	10.83 (0.7)	10.89 (0.3)	10.83 (0.6)	4.95 ± 0.14
**9**	8.39 (0.3)	8.48 (0.1)	8.50 (0.1)	8.46 (0.7)	1.64 ± 0.01

^a^ Values of peak area; relative standard deviations are in the parenthesis.

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
