# Peer review of "Combined MS/MS-NMR Annotation Guided Discovery of Iris lactea var. chinensis Seed as a Source of Viral Neuraminidase Inhibitory Polyphenols"

_molecules, 2020, doi:10.3390/molecules25153383_

Round 1

Reviewer 1 Report

The presented work authors employ variety of analytical and computational methods to characterize and discover stilbene compounds present in Iris lactea seed extract.

I would recommend the publication of this work as it presents very interesting combination of HPLC, MS and NMR with neural network based data analysis.

The research material is very broad that authors had to be concise in methods description to control the paper length. Authors might consider splitting the material into two articles to present experimental work and methods’ conditions in more details.

General comments:

  1. Part 4.2 describing plant materials is not clear. Please describe seed identification and selection process. Also information on length and storage conditions should be included.
  2. The overall sequence of experiments is not entirely clear. Can you summarize this sequence in a form of table or graph also indicating which extract was used in particular experiment. It seems to be several extracts or extracts that underwent MPLC separation. The extraction procedure is described in part 4.3 but there is no correlation with different experiments.
  3. There is no mention if any reference substance was used. No source of reference substance was listed.
  4. Part 4.1 lists equipment for optical rotation and TLC experiments but there is no experiment description nor any results.
  5. For HPLC-UV quantitative method please indicate what substance (which extract) was used for calibration and which as a test substance.
  6. How HPLC-UV specificity was ensured?

Author Response

Dear reviewer

Thank you for your kind advice.

We did our best to correct our manuscript as your advice.

<Suggestion for authors>

The presented work authors employ variety of analytical and computational methods to characterize and discover stilbene compounds present in Iris lactea seed extract.

I would recommend the publication of this work as it presents very interesting combination of HPLC, MS and NMR with neural network based data analysis.

The research material is very broad that authors had to be concise in methods description to control the paper length. Authors might consider splitting the material into two articles to present experimental work and methods’ conditions in more details.

Response: Thanks for your detail, and kind advices, we could edit our manuscript more detail and clear. As shown in a revised manuscript, we described the sample preparation in detailed sections and unified ambiguous expressions to clear expressions.

<General comments>

  1. Part 4.2 describing plant materials is not clear. Please describe seed identification and selection process. Also information on length and storage conditions should be included.

Response: In this revised version, we described the identification of seed [appearance and length (Figure S13), DNA-based analysis (Tables S3-S5)] as below.

“3.2. Plant materials

Dried seeds of Iris lactea var. chinensis were purchased from the oriental herb market, Health Wisdom (https://www.healthwisdom.shop). The dried seeds (Figure S13, Supplementary Materials) of which moisture content was 9.1% used in the present study (SNU-1609) was identified by H.-S. Chae, along with DNA-based authentication (Tables S3-S5, Supplementary Materials) performed by Life Sciences Research Institute, Biomedic Co., Ltd. in South Korea, and deposited at the Medicinal Plant Garden, College of Pharmacy, Seoul National University.”

Because we purchased the plant materials from commercial source, we checked the materials carefully with DNA-based authentication performed by Life Sciences Research Institute, Biomedic Co., Ltd. in South Korea. In the quthentication process, the chloroplast barcoding markers (matK, rbcL, trnLF) were used.

  1. The overall sequence of experiments is not entirely clear. Can you summarize this sequence in a form of table or graph also indicating which extract was used in particular experiment. It seems to be several extracts or extracts that underwent MPLC separation. The extraction procedure is described in part 4.3 but there is no correlation with different experiments.

Response: According to this comment, we adjusted the order of the material and method section based on the experiment steps; extraction, LC-MS analysis, NMR analysis, isolation, quantification, and evaluation of bioactivities.

Further isolation scheme was described in figure S10 of Supplementary Materials. The extracts used for each experiment were also marked on the figure.

Additionally, In the sections 3.4.1. and 3.7.1., sample preparations for each experiment were mentioned.

“3.4.1 Sample preparation

Lyophilized crude extract of I. lactea var. chinensis was dissolved in LC-MS grade MeOH with sonication and filtered with a 0.2 μm PVDF membrane syringe filter (Pall Gelman Sciences, Ann Arbor, MI) with a concentration of 2 mg/mL.”

“3.7.1. Preparation of standard and sample solution

Three isolated and purified compounds, trans-ε-viniferin (3), vitisin A (6), and vitisin B (9) were used as reference compounds. Stock solutions (10 mg/mL) were prepared by precisely weighting and dissolving each in 50% aqueous methanol (v/v) before every experiment day. A mixed standard stock solution was prepared by adding precise volume of each stock solution to make a solution containing 1 mg/mL of each analyte. The mixed solution was diluted via serial dilution. The crude extract power (100 mg) was precisely weighted and dissolved in 50 % aqueous methanol (v/v) with sonication before every experiment day, and the solution was diluted to sample solution concentration (3.33 mg/mL). All solutions were filtered through a 0.2 μm PVDF membrane filter before injection.”

In SMART analysis, we used the ethyl acetate extract belonging to general extraction process, so the expression, ‘secondary metabolites enriched fraction’ was changed to ‘ethyl acetate extract of I. lactea var. chinensis’.

  1. There is no mention if any reference substance was used. No source of reference substance was listed.

Response: There were no available commercial sources for vitisins A and B, we used all isolated compounds as reference compounds. All compounds were carefully identified by analysis of NMR and MS data. Chromatograms of each compound were in figure S11. This information was also described in section 3.7.1. on the revised manuscript.

  1. Part 4.1 lists equipment for optical rotation and TLC experiments but there is no experiment description nor any results.

Response: Optical rotation data were measured and reported in section 3.6. TLC data were not described on the manuscript, but we used the TLC experiments basically during the isolation including Sephadex LH-20 column chromatography and MPLC experiments. Thus, we described the specification of TLC plates which we used in this experiment.

  1. For HPLC-UV quantitative method please indicate what substance (which extract) was used for calibration and which as a test substance.

Response: We described the detailed information of sample solution in section 3.7.1. as below

“The crude extract power (100 mg) was precisely weighted and dissolved in 50 % aqueous methanol (v/v) with sonication before every experiment day, and the solution was diluted to sample solution concentration (3.33 mg/mL).”

  1. How HPLC-UV specificity was ensured?

Response: We compared the UV absorbance patterns from each reference compound and each peak from sample solution (Figure S12) to ensure the specificity. From the result, the UV absorbance patterns were found to be identical to each other.

We sincerely appreciate your consideration and hope that our responses will be appropriate to your comments.

Reviewer 2 Report

I believe that the article is a very interesting one. However, the authors should better underline the novelty of their study in the introduction part. Also the authors did not respect the journal format. The conclusions part must be the final one to the article and not before the materials and methods part.

Author Response

Dear reviewer

Thank you for your kind advice.

We did our best to correct our manuscript as your advice.

<Suggestion for authors>

I believe that the article is a very interesting one. However, the authors should better underline the novelty of their study in the introduction part. Also the authors did not respect the journal format. The conclusions part must be the final one to the article and not before the materials and methods part.

Response: Thanks for your precious comments. According to this comment, authors added a new paragraph in the introduction as below.

“Mass spectrometry and NMR spectroscopy are the most common techniques in metabolomics studies, and each brings its strength and weakness. In mass spectrometry, the structure information was described by the size of molecular ion, and the fragmentation patterns of the molecular ions with high sensitivity. The diversity of fragmentation ion patterns belongs to molecular structures, so analysis of these patterns. However, results from mass spectrometry depends on the ionizabilities of targeted molecules. Unlike mass spectrometry, NMR spectroscopy is quantitative and produces highly reproductive results. Even the sensitivity of NMR is lower than that of mass spectrometry, NMR data is not influenced by the type of molecules. Moreover, the sensitivity and acquisition time have been improved by development of cryo- and microprobes [32]. Thus, complementary application of these two techniques is appropriate to study natural products which have a plenty of chemical diversity.”

The conclusion part was located at the final position and the formatting was checked again.

We sincerely appreciate your consideration and hope that our responses will be appropriate to your comments.

Reviewer 3 Report

Minor remarks

Avoid the first-person plural (we) and only use the third-person singular in the scientific paper.

Line 27… provide indefinite article “a” between “oligomers with” and “human health benefit”.

Line 32…Instead of “are able to” use “can”.

Line 33…Provide “-” between “plant” and “derived”.

Line 39…Insert comma after “hepatoprotective”.

Lines 40 and 41…”L.” should be given using normal letter (not italic letters).

Line 44…delete “moreover,”

Line 45…insert comma after “Polygonaceae”.

Line 50…delete “the purpose of”.

Line 52…delete “so as”.

Line 53…insert “the” between “containing” and “new”

Line 56… provide blank “analysis,SMART”

Line 59…delete “the” in “provide the useful”.

Line 60…should be inserted “discovery of a similar type of structure”

Line 64…should be inserted “on the website”.

Line 65…should be inserted “allows characterizing the natural”

Line 87…should be inserted “none of the nodes”

Line 95…should be typed “each of the nodes”

Line 139 and Line 206… Instead of “In addition” use “Also,”

Line 169…should be inserted “configuration of the aglycone part”.

Line 194…Latin name should be presented using italic letters.

Line 207…insert comma after “Further”.

Line 207…use “dose-dependent”.

Line 208…”50” in “IC50” should be indexed.

Line 210…use “However, a previous study”.

Line 216…Instead of “was” use “were”.

Line 218…use ”in the last stage”.

Line 219…use “in early-stage within”.

Line 223…use “seeds as a source of”

Line 240…use “with a cryogenic”.

Line 262… Through the manuscript, the blank should be inserted between quantity and unit, except in the case of percentage. Check this through the whole paper. Also, provide blank between quantity and unit “(3×1.5L, 7.1 g)”.

In Conclusion, use italic letters for Vitis

Line 270...”-1” in “40 mL min-1” should be placed in a superscript.

Line 270, Line 271, Line 273, Line 275…use “flow rate of”

Line 282 and Line 285…provide full stop at the end of paragraph.

Line 288, Line 292…delete blank between “auto sampler”

Line 299… m/z 554.2615 is desirable to be rounded.

Line 301… insert “using a feature-based”

Line 310…use “phase-corrected”

Line 341…instead of “mins” use SI unit for time, “min”. The plural for the unit should be avoided.

Line 349…after NMR insert comma.

Line 357…after Kim insert comma.

Line 383…provide full stop in the abbreviated title of the journal.

Line 385…provide pages in the range of 265-278.

Line 401…provide pages in the range of 144-168.

Line 427…delete “1” after “Vitis coignetiae”.

Line 441…use the following abbreviated title of the journal (BMC Bioinform.)

In Figure 3, provide the labels and units for abscissa and ordinate.

In Table 1, m/z is desirable to be rounded to an integer as well as through the paper (for instance, Line 128 and Line 155).

In section 4.3…be specific when you mention the percentage of ethanol. You probably mean the volumetric percentage. This information should be provided in this part of the manuscript.

In supplementary, delete highlight in Figure S3 and provide the labels and units for abscissa and ordinate.

Major remarks

The moisture content of plant material should be determined and provided in Materials and Methods.

The references should be excluded from the conclusion. This part should be retyped and also reduced.

If possible, replace older references with newer ones.

Author Response

Dear reviewer

Thank you for your kind advice.

We did our best to correct our manuscript as your advice.

<Suggestion for authors>

Thanks to your detail advices, we corrected all expression as your notice.

Mionor corrections were made at this time according to your points.

In Figure 3, provide the labels and units for abscissa and ordinate

Response: Relative abundance (%) for figure 3(A) and absorbance (mAU) for figure 3(B) were provided at this revised version.

In Table 1, m/z is desirable to be rounded to an integer as well as through the paper (for instance, Line 128 and Line 155).

Response: Thanks to your advice. However, we remained the m/z of precursor ion 4 digits values because all 4 digits values were measured by high resolution mass spectrometry and we believed these data would be help to other researchers.

In section 4.3…be specific when you mention the percentage of ethanol. You probably mean the volumetric percentage. This information should be provided in this part of the manuscript.

Response: We totally agreed with your advices. We corrected all expression involved with volumetric percentage by adding v/v.

In supplementary, delete highlight in Figure S3 and provide the labels and units for abscissa and ordinate.

Response: Correction was made.

Figure S3. ECD spectrum of compound 1

Major remarks

The moisture content of plant material should be determined and provided in Materials and Methods.

Response: We agreed with your suggestion. We measured the moisture contents of I. lactea var. chinensis seeds using loss on drying method. The moisture content was measured as 9.1% (w/w) and we mentioned this result in the plant material section on the revised manuscript. Consequently, the contents of stilbene oligomers were also corrected based on this result.

“The dried seeds (Figure S13, Supporting Information) of which moisture content was 9.1% used in the present study (SNU-1609)”

The moisture content was measured by the following method:

The sample (2.0847 g) was carefully weighted and dried at 105℃ over 6 h. The dried sample was carefully moved to a desiccator to avoid moisturization and cooled down. The dried sample (1.8950 g) was carefully weighted to measure the loss (0.1897 g). Finally, the moisture content was calculated to 9.1%.

The references should be excluded from the conclusion. This part should be retyped and also reduced. If possible, replace older references with newer ones.

Response: According to this comment, we removed the reference from the conclusion.

Also, we changed the reference 22 and 25 to newer ones to provide NMR data.

Reference 22 (“Tetrahedron Lett. 1998, 54, 6651-6660”) has been changed to “Phytochem. Anal. 2019, 30, 320-331”.

Reference 25(“Tetrahedron 1995, 51, 11979-11986”) has been changed to “Chem. Pharm. Bull. (Tokyo) 2007, 55, 899-901”.

We sincerely appreciate your consideration and hope that our responses will be appropriate to your comments.

Round 2

Reviewer 3 Report

Minor remarks

Line 152…provide “in” before “refrigerator” and also provide a full stop at the end of the sentence.

Line 221…should be inserted as follows: “in 50% (v/v) aqueous methanol”

Line 224…should be inserted as follows: “50% (v/v) aqueous methanol”. It is necessary to delete a blank between quantity and unit and (v/v) should be placed close to %.